# In Silico and In Vitro Structure–Activity Relationship of Mastoparan and Its Analogs

**DOI:** 10.3390/molecules27020561

**Published:** 2022-01-16

**Authors:** Prapenpuksiri Rungsa, Steve Peigneur, Nisachon Jangpromma, Sompong Klaynongsruang, Jan Tytgat, Sakda Daduang

**Affiliations:** 1Protein and Proteomics Research Center for Commercial and Industrial Purposes (ProCCI), Khon Kaen University, Khon Kaen 40002, Thailand; prapenpuksiri@gmail.com (P.R.); nisaja@kku.ac.th (N.J.); somkly@kku.ac.th (S.K.); 2Division of Pharmacognosy and Toxicology, Faculty of Pharmaceutical Sciences, Khon Kaen University, Khon Kaen 40002, Thailand; 3Toxicology and Pharmacology, Campus Gasthuisberg, University of Leuven (KU Leuven), O&N 2, P.O. Box 922, Herestraat 49, 3000 Leuven, Belgium; steve.peigneur@kuleuven.be; 4Department of Biochemistry, Faculty of Science, Khon Kaen University, Khon Kaen 40002, Thailand

**Keywords:** mastoparan, antimicrobial peptides, hemolysis, circular dichroism, venom, wasp

## Abstract

Antimicrobial peptides are an important class of therapeutic agent used against a wide range of pathogens such as Gram-negative and Gram-positive bacteria, fungi, and viruses. Mastoparan (MpVT) is an α-helix and amphipathic tetradecapeptide obtained from *Vespa tropica* venom. This peptide exhibits antibacterial activity. In this work, we investigate the effect of amino acid substitutions and deletion of the first three C-terminal residues on the structure–activity relationship. In this in silico study, the predicted structure of MpVT and its analog have characteristic features of linear cationic peptides rich in hydrophobic and basic amino acids without disulfide bonds. The secondary structure and the biological activity of six designed analogs are studied. The biological activity assays show that the substitution of phenylalanine (MpVT1) results in a higher antibacterial activity than that of MpVT without increasing toxicity. The analogs with the first three deleted C-terminal residues showed decreased antibacterial and hemolytic activity. The CD (circular dichroism) spectra of these peptides show a high content α-helical conformation in the presence of 40% 2,2,2-trifluoroethanol (TFE). In conclusion, the first three C-terminal deletions reduced the length of the α-helix, explaining the decreased biological activity. MpVTs show that the hemolytic activity of mastoparan is correlated to mean hydrophobicity and mean hydrophobic moment. The position and spatial arrangement of specific hydrophobic residues on the non-polar face of α-helical AMPs may be crucial for the interaction of AMPs with cell membranes.

## 1. Introduction

In recent years, the development of multidrug-resistant bacteria, in particular in hospital patients, has led to an increased interest in the search for new antibiotics [1]. The search for new antibiotic substances found an almost inexhaustible source of potential therapeutic agents amongst antimicrobial peptides (AMPs), with sources including bacteria, archaea, protists, fungi, animal venom, and plants [2,3]. AMPs are usually small (lower than 100 amino acids), charged, amphipathic molecules found as components of an organism’s innate immune system, and have a broad-spectrum antimicrobial activity or are involved in immune regulation, wound healing, or apoptosis. AMPs can resemble a wide variety of structures but typically have an excess positive charge of +2 to +9, around 50% hydrophobicity, and the acquisition of a different secondary structure in membranous environments [4]. In the Antimicrobial Peptide Database (APD), 14% of the antimicrobial peptides known to date adopt an α-helix structure [5]. In general, the degree of helicity and amphipathicity seems to be important for antimicrobial efficiency [6]. Social Hymenoptera represents the group of most venomous insects that sting, such as bees, wasps, and ants [7,8,9,10]. Their venoms are a complex mixture of biologically and pharmacology active components such as biogenic amine, peptides, and proteins [11,12] that are natural chemical weapons evolved to be used for defense, predation, and protection against pathogenic microorganisms [7,13]. Mastoparan is a 14-amino acid, amphipathic, and cationic peptide obtained from wasp venom [14,15,16,17]. They exhibit several biological activities, including mast cell degranulation, hemolysis, and the release of histamines [15]. Their activity against both Gram-positive and Gram-negative bacteria and yeast has been reported [9,15,18]. Most mastoparan sequences have many hydrophobic amino acids and 2–4 Lys residues. The majority of hydrophobic nature amino acids in mastoparans peptides leads them to adopt an amphipathic α-helix conformation in micelle solvents and membrane-bound state [19]. The physicochemical properties of the peptides, such as the length, charge distribution, net charge, molecular volume, amphipathicity, and oligomeric state in solution, play essential roles in the interactions between the peptide and the membrane surface and/or with the membrane core [20]. The wasp venom mastoparan represents a potential source of pharmacology substances AMPs. However, some of the wasp venom mastoparans are not optimal for therapeutic applications due to limitations such as toxicity, salt sensitivity, and bioavailability, which limit their evolution into therapeutically effective antibiotics.

In this regard, various approaches have been employed to optimize the original sequence of AMPs, including residue substitution and/or truncation. Mastoparans have inspired the development of novel antimicrobial agents, focusing on increasing antimicrobial activity compared to the natural peptides. However, natural mastoparan has been reported to have high toxicity effects and hemolytic activity. In addition, developed mastoparans are similar to natural AMPs, including restricted half-lives, stability in addition to potential, and reduced toxicity. Furthermore, there have been efforts to develop potent antimicrobial peptides by modifying their amino acid sequences. The peptides’ structure, changed by amino acid substitution, deletion, truncation, and hybridization, was influenced by its physiological properties [1,19,21].

In this work, the structural and functional characterization of mastoparan peptide was identified in Thai Banded wasp, *Vespa tropica*. A series of synthetic *V. tropica* mastoparan (MpVT) peptides was designed and their structure–activity relationships were investigated. We demonstrate that the engineering of MpVT is a promising strategy for the development of subtype-selective therapeutics with an interesting antimicrobial activity.

## 2. Results

### 2.1. Purified Peptides and Peptides Synthesis

Soluble *V. tropica* crude venom was fractionated by reverse-phase HPLC, and the antimicrobial activity of the fractions was investigated. The chromatographic profile of the venom is shown in the Appendix A. The antimicrobial activity fraction was determined by N-terminal sequences by automated Edman degradation using a Shimadzu PPSQ-30 protein sequencer [22].

The following sequence was obtained INLKAIAAFAKKLI/L. A homology search using the BLAST search engine tool (http://www.ncbi.nlh.gov/BLAST/) accessed on 25 June 2021 indicated that these peptides are homologous to mastoparan peptides named MpVT. To investigate the key residues for antimicrobial activity, we synthesized six analogs. The helical wheel of MpVT was used to segregate the hydrophilic and hydrophobic residues, which led to new peptides that were composed of an amphipathic α-helix. Table 1 shows the sequences of these peptides, their respective net charges, and the values of mean hydrophobicity and hydrophobic moment, calculated according to the consensus scale of Eisenberg et al. [23].

### 2.2. The Partial cDNA Cloning and Sequence Comparison

The cDNA sequences encoding MpVT were identified. The cDNA sequencing resulted in a sequence composed of an anionic pro sequence and a mature peptide sequence with an additional glycine at the C-terminus (Figure 1). The partial MpVT nucleotide sequences contain 241 bp consisting of a pro sequence of 111 bp, a mature 45 bp, and 3′ UTR 85 bp (Figure 1A). As shown in Figure 1B, the deduced amino acid pro sequences were encoded by 111 bp in length, consisting of 22 amino acid residues. The MpVT mature protein contains 15 amino acid residues corresponding to the purified and sequenced fraction (^1^INLKAIAAFAKKLI/L^14^). The mature peptides contain three lysines at positions 4, 11, and 12 (Figure 1). However, at the end of the mature sequence, glycine serves as the –NH^+^_4_ donor for the amidation of the C-terminal leucine, similar to what was observed for many other small bioactive peptides such as other AMPs and bombasin [12,17,24].

### 2.3. Physicochemical Properties and Secondary Structure of MpVT and Its Analogs

The design of mutant peptides by amino acid substitution and deletion led to an effect on their physicochemical properties. The most hydrophobic amino acid residues, i.e., Leu, Ile, Ala, Val, Phe, Met, and Trp, are located on one side, whereas the hydrophilic amino acid residues Lys, Asn, and Gly are located on the other side of the α-helix. The substitution of Phe^13^ (MpVT1) and Ala^12^ (MpVT3) did not change the angle of the helix face (Figure 2B,C). Only for MpVT3, the total net charge changed from +3 to +2 because of the Lys change to Ala.

The secondary structure prediction of MpVT and its analogs shows that these peptides form an amphipathic α-helix conformation under different environments. The MpVT was mapped into a helical wheel diagram, and the analogs were designed by specific amino acid residue substitutions and deletions (Figure 2A–G).

The first three amino acids of MpVT were deleted in MpVT4-MpVT7. The hydrophilic amino acids of MpVT4, MpVT5, MpVT6, and MpVT7 (Asn) were changed to hydrophobic amino acid (Ala/Phe), which caused a change in the helix face and decreased the net charge from +3 to +2. The physicochemical properties of all peptides are summarized in Table 2.

### 2.4. The Secondary Structure Evaluation

The secondary structures of the MpVT and its analogs were determined by CD spectroscopy in the presence of water and membrane mimic conditions and the presence of 40% 2,2,2-Trifluoroethanol aqueous solution (TFE) (Figure 3A–G). CD spectra of MpVT and its analog in water confirmed an unordered conformation. However, in the presence of 40% TFE, a strong positive dichroic band near 192 nm and two negative bands around 208 nm and 222 nm are observed, indicating an α-helix conformation. The deconvolution of the spectrum was determined using the K2D3 program [25].

The CD spectra of MpVT, MpVT1, and MpVT3 were similar, indicating that in these environments, a higher helical content was induced that was more hydrophobic (Table 2), representing a spectrum compatible with the induced an α-helix structure. The spectra of MpVT4–7 show typical random coil structures, reflecting poor binding to the membrane mimetic system (Figure 3D–G).

### 2.5. Structural Analysis

To obtain insights into the three-dimensional structure of MpVT and its analogs, molecular modeling simulations were performed (Figure 4A–G). The amino acid substitution of MpVT was designed into MpVT1 and MpVT3 (Table 1). The replacement of Leu^13^ to Phe^13^ in MpVT1 increased the antimicrobial activity, as shown in Figure 4B.

This substitution resulted in a higher antimicrobial activity against the *E. coli* strain. In MpVT3, the positive amino acid Lys was replaced by the non-polar amino acid (Ala), as shown in Figure 4C.

The truncation of the amino acid from the N-terminus of the MpVT peptide chain (MpVT4, MpVT5, MpVT6, and MpVT7), in which the first three MpVT amino acids (^1^INL^3^) was deleted (Table 1), caused a gradual decrease in antimicrobial potency against both bacterial strains (Table 3). A gradual truncation of the peptide chain at the N-terminus resulted in a steeper decrease in antimicrobial activity. Moreover, it is possible to reduce the length of the α-helical AMPs (Figure 4D–G) without affecting their antimicrobial activity.

### 2.6. Antibacterial Activity

The biological activities of MpVT and its analogs were investigated by assaying antimicrobial activities and by studying the hemolysis activity. The minimal inhibitory concentrations (MIC) of the peptides were determined based on methods described by Anunthawan et al. [26]. Table 3 shows the results of the antibiotic assays against Gram-positive and Gram-negative bacteria. The peptides were highly active against the Gram-negative bacteria. MpVT1, especially, showed the highest potency of all MpVT peptides tested (Figure 5A–F). The MpVT4-7 could not detect antibacterial activity when using a peptide concentration lower than 50 μg/mL. However, the *E. coli* DH5α were highly sensitive to antibacterial activity, which is shown in Table 2. The *E. coli* strains that were the most susceptible against MpVT peptides were selected for visualizing the bacterial morphology when treated with these peptides using a scanning electron microscope [27]. The normal and abnormal morphologies of bacteria cells were visualized after 60 min treatment with the MpVT peptides (Figure 6).

### 2.7. Time–Kill Assay

Regarding the time course of antimicrobial activity of MpVT, MpVT1, and MpVT3 able to kill *E. coli* 0157:H7 at 2× MIC, the time curves reveal that MpVT1 and MpVT3 caused a rapid, complete killing of *E. coli* 0157:H7 within 60 min, and a slower killing MpVT within 90 min (Figure 7). These results indicate the bactericidal effects of MpVT1 and MpVT3 on these bacterial strains with a higher susceptibility of the Gram-negative *E. coli* compared with the peptide.

### 2.8. Hemolytic Activity

The hemolytic activity of MpVT and its analog were investigated on human red blood cells (hRBCs). Figure 8 shows that all peptides exert a hemolysis effect on hRBCs of around less than 0.5% hemolysis compared with a 100% hemolysis of 0.1% (*v*/*v*) Triton X-100 as a positive control. According to Figure 8, MpVT3, MpVT1, MpVT6, and MpVT7 present the highest hemolytic activity of peptides, respectively. However, MpVT and its analogs show weak hemolytic activity, less than 10% at 50 μg/mL. In addition, only MpVT3 exhibits hemolytic activity, more than 50% against hRBCs at 100 μg/mL.

## 3. Discussion

The venom produced by venomous creatures, including snakes, jellyfish, spiders, and scorpions, as well as some insects, such as bees or wasps, has evolved because of their direct benefits to these organisms in terms of self-defense and/or prey acquisition [28]. Many bioactive substances from the venom often also possess strong antimicrobial and anti-cancer properties [17,29,30]. The biological activity of AMP could be affected by many structural parameters, including size, sequence, charge, hydrophobicity, hydrophobic moment, and conformation [16].

Stings of social wasps from the genus Vespa are medically important and known to cause a variety of allergic reactions [31] and direct toxic effects [32,33]. Cases in which multiple stings are received can be clinically threatening, often resulting in acute renal failure, a difficult clinical situation usually requiring long-term hospitalization. Protecting the kidneys is the first and most common concern when dealing with massive wasp sting envenomation [34]. The great banded wasps (*V. tropica*) are commonly from in the Northeastern part of Thailand. Their venoms are highly potent against several animals. Their nests are often built in close proximity to human settlements, resulting in many cases of accidental stings annually [7,35].

The sequence comparison of MpVT with sequences in the available databases shows that these peptides are similar to mastoparan-VT and other mastoparan peptides [12,36]. Mastoparans represent the most abundant class of peptides in the venom of wasps. They are rich in hydrophobic and basic amino acids and do not contain disulfide bonds. They have the characteristic feature of short, cationic linear α-helical amphipathic peptides with 10 to 14 amino acid residues [21,29,37]. Puri and Roche (2008) reported the potential for mastoparan peptides to cause cytolytic action in animal cells [6], and these peptides also play important roles in allergy and inflammation because they activate exocytosis and granule fusion with the plasma membrane, which, as a consequence, releases histamine, proteases, lipid mediators, and cytokines [38]. The MpVT sequences are composed of an *N*-terminal signal sequence, an anionic prosequence, and a mature peptide sequence with an appendix glycine. The first isoleucine position of the mature sequences is commonly found in vespid venom [15,16,39]. Through post-translational modification, including cleavage of a signal sequence, removal of prosequence, and C-terminal amidation, the precursor polypeptides are processed to form mature mastoparans [16]. The C-terminal amidation of mastoparans was found to be a common feature in hornet and wasp species [15,40]. It stabilizes the α-helical conformation by providing an extra hydrogen bond [41]. Moreover, the amidated mastoparan peptides and other antimicrobial peptides show enhanced antimicrobial activity in comparison with the deamidated peptides [12,31]. Based on the *N*-terminal sequences, the lasted amino acid Ile^14^ was able to Leu^14^, which is an isomer amino acid. The Leu^14^ sequence corresponded to deduced cDNA sequences [17].

MpVT was designed based on multiple active peptides and synthesized to investigate the residues important for activity. The structure–activity relationship of MpVT and MpVT1-3 analogs was determined by verifying the number and positions of the Ala/Phe residues. The results show that the substitutions also cause simultaneous changes in the mean hydrophobicity of the peptides as well as in other structural parameters, such as the hydrophobic moment, the size of the polar face, and the overall amphipathicity. The α-helical AMPs have highly potent antimicrobial properties that require several structural and physicochemical parameters such as size, net charge, hydrophobicity, amphipathicity, and helical propensity [1]. According to our results, the highest bactericidal activity of MpVT1, MpVT3, and MpVT could be distributed to the perfected structure. Furthermore, the “Leu^13^” of MpVT was changed to “Phe^13^”, which enhanced its hydrophobic moment and improved the antibacterial activity in MpVT1 peptides. Corresponding to some studies, the quantitative measure of amphipathicity is the hydrophobic moment (μH), which might be the most important parameter governing antimicrobial activity [42,43]. Based on the predicted structure of the designed AMPs, the position and spatial arrangement of the specific hydrophobic residues such as the bulky amino acid (Phe, Tyr, or Try) on the non-polar face of α-helical AMPs may be crucial for the interaction of AMPs with the cell membrane [44]. In AMP sequences, effective pinning of a large number of peptides charges the polar headgroup region of the membranes, which, in turn, facilitates membrane defect formation and rupture. Due to the large size of the bulky amino acid group (Phe, Tyr, or Try), combined with their surface localization, part of the selectivity between bacterial and eukaryotic cell membranes is obtained through cholesterol precluding membrane insertion of the W/F groups and through lower adsorption of the cationic composite peptides at zwitterionic than at anionic membranes [27]. Deletion of the first three amino acids of MpVT (^1^INL^3^) caused a gradual decrease in antimicrobial potency against both bacterial strains. These results correspond to a previous report in which truncation at the N- or C-terminal of the peptides chain caused a gradual decrease in antimicrobial potency [44]. This corresponds to Silva et al.’s 2014 description of the mechanism of action of mastoparan analogs, in which both the N- and C-terminal remain positioned outside the membrane, while the α-carbon backbone becomes partially embedded in the membrane core, assuming the pore-forming mode of action.

Killing kinetics assay of MpVTs were performed two-times of MIC against *E. coli* 0157:H7; the results show that MpVT1 and MpVT3 achieved a complete killing within 60 min. However, MpVT was shown to complete killing at 90 min. This property indicates that there is less chance of microbes developing resistance to it. The case of antibiotic resistance is often discovered in bacteria exhibiting a long annihilation time with a specific antibiotic, consequently allowing strains sufficient time to revert themselves.

Circular dichroism spectroscopy analysis of MpVT and three analogs in a different environment show that all these peptides experience important conformational changes from a random coil conformation in a water solution to a helical structure that is induced by the electrically neutral 40% TEF solution induced in peptides. These results suggest that hydrophobic interactions are the most important since they induce conformational changes and are important for binding to the membrane surface. The short polycationic peptides often assume helical conformations, and the degree of α-helix is directly related to the burial of the backbone into a more hydrophobic region and herewith to their ability to disturb the membranes [45].

Circular dichroism analysis in different environments showed that these peptides undergo important conformational changes from a random coil in an aqueous solution to helical structures induced by anisotropic environments. The results reported above made clear that upon binding to the membrane surface, the short polycationic peptides often assume helical conformations, and the degree of α-helix is directly related to the burial of the backbone into a more hydrophobic region and to their ability to disturb the membranes. The anionic micellar environment (SDS 8 mM) and the electrically neutral 40% (*v*/*v*) TFE solution induced approximately the same amount of secondary structures in the three natural peptides, suggesting that hydrophobic interactions may be more important than the electrostatic in inducing these conformational changes.

Since MpVT and its analogs belong to the mastoparan family, they may have hemolysis activity, which is a problem for their clinical use, as they are capable of destroying eukaryotic cells. The hemolysis assay was used to evaluate the cytotoxicity of MpVT and its analog molecules against human red blood cell concentrate [6,29]. At 100 μg/mL, the MpVT and its analogs have hemolysis activity less than 50%. However, MpVT3 has hemolysis activity of more than 50%; these results may cause Lys^12^ to change to Ala^12^. The hemolytic activity seems to be directly dependent on membrane perturbation caused by the peptides due to their interaction with the zwitterionic membrane of the erythrocytes. The most active hemolytic peptides were MpVT3, MpVT1, MpVT7, and MpVT6. All of these peptides present the lower charge (+2) except for MpVT1 (+3).

The capacity to exert hemolytic activity of peptides is directly related to interaction with zwitterionic membranes of the red blood cells. Some of the characteristics of a peptide that can contribute to this effect are the hydrophobic nature of most amino acid residues in chains and their organization in amphipathic α-helical conformation [15]. The electrostatic interactions between positive residues of peptides and negative charge of phospholipid head groups present in biological membranes, which facilitate peptide insertion, leading to destabilization and consequent lysis.

Some studies have reported that the net charge of peptides significantly affects hemolytic activity, in which a decrease to a level lower than +4 can make the peptide totally inactive, but the systematic increase in this property from +4 to +8 can make the peptide exhibit hemolytic activity. The verification of MpVT and its analogs showed the net charge at pH7 is less than +4.

In conclusion, amphipathicity, hydrophobicity, and net positive charge are important parameters that modulate the antimicrobial activity of α-helix structure peptides.

## 4. Materials and Methods

### 4.1. Peptides Synthesis and Purification

In total, 10 mg of the freeze-dried total venom was purified using high-performance liquid chromatography (HPLC) using a C18 reversed-phase column (dimensions 4.6 × 250 mm, 5 μm Vydac 218 MS C18), as previously reported [22]. A Shimadzu PPSQ-30A protein sequencer was used to determine the *N*-terminal amino acid sequences of the *V. tropica* mastoparan (MpVT) peptide. The peptides were synthesized by GenicBio Limited (Shanghai, China). The MpVT was used as the template for seven peptide modifications. Their analogs were named MpVT1-MpVT6, and their sequences are shown in Table 1.

### 4.2. cDNA Cloning and Sequence Comparison

*V. tropica* total RNA was extracted from the venom gland using TRIzol^®^ reagent (Invitrogen, Waltham, MA, USA), as described by Rungsa et al. (2018) [22]. RT-PCR was applied according to a protocol of the RevertAid First Strand cDNA synthesis kit (Thermo Scientific, Waltham, MA, USA). The gene-specific primers used were following a previously reported conserved region of the Vespa spp. mastoparan by Lin et al. (2011) [16]. The PCR condition was set up as follows: initial denaturation at 95 °C for 5 min, followed by 35 cycles of denaturation at 95 °C for 30 s, annealing at 55 °C for 30 s, elongation at 72 °C for 2 min, and final elongation at 72 °C for 10 min. The PCR products were cloned into the pGEM^®^-T easy vector (Promega, Madison, WI, USA) for sequencing [46].

### 4.3. Molecular Modeling

The short antimicrobial peptides of MpVT and its analog in this study are shown in Table 1. The primary sequences of MpVT were searched, and alignment was produced with other mastoparan peptides using PDB at www.uniprot.org/uniport/, accessed on 1 May 2021. The structure prediction of short peptides (10–30 amino acids) was initially predicted using online software PEP-FOLD [47,48]. The model was created by using 3D-HM: the 3D Hydrophobic Moment Vector Calculator at http://www.ibg.kit.edu/HM/?page=helix, accessed on 1 May 2021. The best models were evaluated through PROCHECK. PROCHECK checks the stereochemical quality of a protein structure through the Ramachandran plot, in which good quality models are expected to have 90% amino acid residues in the most favored and additionally allowed regions. Structure visualization was performed in UCSF Chimera at https://www.cgl.ucsf.edu/chimera/, accessed on 1 May 2021. Helical structure analysis was performed using Netwheels: Peptides Helical Wheel and Net projections maker at http://lbqp.unb.br/NetWheels/, accessed on 1 May 2021 and confirmed wheels using heliQuest at https://heliquest.ipmc.cnrs.fr/cgi-bin/ComputParams.py, accessed at 1 May 2021. The peptides’ properties were calculated following the Eisenberg, et al. (1982) scale [23].

### 4.4. Determination of Minimum Inhibitory Concentration (MIC)

The microorganisms; *Staphylococcus auerus* ATCC 27,853 (DMST 4739), *Staphylococcus auerus* ATCC 25923, *Bacillus subtilis* ATCC663, *Escherichia coli* 0157:H7, *Escherichia coli* DH5α, and *Klebsiella pneumoniae* ATCC27736 (DMST4739) were used to determine the MIC of MpVT and its analogs. The bacterial strains were inoculated to enter their log phases, and then the suspension was diluted in nutrient broth. The microorganisms were prepared at a final concentration of 1 × 10^5^ CFU/mL. The peptides’ concentrations, ranging from 0.1 to 50 μg/mL, were determined. Serial doubling dilutions of various concentrations of MpVTs were made into a nutrient broth to a final volume of 100 μL in each well of a 96-well plate.
A% inhibition = 100 − (O.D._600 at 18 h_ − O.D._600 at immediate treatment_ × 100/O.D._600 at 24 h of control_ − O.D._600 at 18 h of control_ −O.D._600 at immediate treatment of control_). (1)

The MIC is defined as the concentration in which the growth inhibition was greater than 90% of the MpVTs that completely inhibit bacterial growth [26].

### 4.5. Hemolysis Activity

To determine the toxicity of the peptides to normal mammalian cells, hemolytic activity assay was used. The hemolytic activity of MpVT and its analogs was determined via the evaluation of hemolysis of human red blood cells (hRBCs) following Theansungnoen et al. (2016) [27]. Briefly, 2.5% (*v*/*v*) hRBCs suspension was prepared in phosphate-buffered saline (PBS), pH 7.4, and 100 μL of suspension was mixed gently with various peptide concentrations, incubated at 37 °C for 1 h. The samples were then centrifuged, and the absorbances of the supernatants were measured at 415 nm. The absorbance measured from lysed hRBCs in the presence of 1% (*v*/*v*) Triton X-100 was considered to be 100%. Hemolysis (%) was calculated using the following equation:hemolysis (%) = [(A_MpVT_ − A_PBS_)/(A_0.1% Triton X-100_ − A_PBS_)] × 100%. (2)

Results are expressed as triplicated.

### 4.6. Secondary Structure Evaluation through Circular Dichroism (CD)

Circular dichroism was performed on a Jasco J-815 CD spectrometer (JASCO International Co., Ltd., Tokyo, Japan). The peptides were resuspended in a different environment with buffer water, 40% 2,2,2-trifluoroethanol aqueous solution (TFE, Merck, Darmstadt, Germany). CD spectra were recorded from 190–260 nm, averaging six scans, and collected at 25 °C using the quartz temperature-controlled cells with a path length of 0.1 cm. Data were recorded at a scan speed of 20 nm/min, bandwidth of 1.0 nm, 1 s response, and 0.1 nm resolution. The percentage of the α-helix structure was predicted via the K2D method on the *DichroWeb* website (http://dichroweb.cryst.bbk.ac.uk/html/home.shtml, accessed at 25 June 2021) using the method following Souza et al. (2005) [49].

### 4.7. Scanning Electron Microscopy (SEM)

The effects of MpVT and its analogs on bacterial cells were monitored under a scanning electron microscope with a slight modification of the method of Theansungnoen et al. [27]. Briefly, *E. coli* 0157:H7 were separately cultured in a nutrient broth to log phase and then centrifuged at 3000× *g* for 5 min. Cell pellets of bacteria were washed twice with 10 mM sodium phosphate buffer, a pH 7.2, and resuspended with the same buffer to a final concentration of 1 × 108 CFU/mL^−1^. Aliquots of the bacterial suspensions were incubated with a five-fold MIC of peptide sample at 37 °C for 60 min. After that, 100 μL of the bacterial solution was carefully applied on a 0.2 μm cellulose acetate membrane (Sartorius AG, Darmstadt, Germany) for 30 min; then, these cells were attached with 300 μL of 2.5% (*v*/*v*) glutaraldehyde solution (Sigma, Madison, WI, USA) for 1 h. The attached cells were dehydrated by subsequently rinsing with a series of ethanol solutions, including 30, 50, 70, 90, and 100% (*v*/*v*) ethanol solutions. The membranes of the bacterial samples were individually attached to a sputter coater (SC7620, Polaron, Cambridge, UK) and then coated with gold–palladium. The antimicrobial effects on each coated specimen were monitored under a scanning electron microscope (LEO1450VP, LEO Electron Microscopy, Cambridge, UK) operating at 12–20 kV. Untreated bacterial cells were used as a control.

### 4.8. Time–Kill Assay

The antimicrobial effect of MpVT and its analogs with a given reaction time was investigated. The *E. coli* 0157:H7 were inoculated in the mid-log phase and then washed three times with 10 mM sodium phosphate buffer pH7.4. The 2× MIC concentration of peptide solution was incubated with a bacteria suspension (1 × 106 CFU/mL). After 0, 15, 30, 60, and 90 min, an equal volume of samples was withdrawn, serially diluted in NB, and spread uniformly onto agar plates. Bacteria colonies on plates were counted after incubation at 37 °C for 24 h. Percent survival was calculated as (T_90_/T_0_) × 100, where T_90_ and T_0_ represent the colony-forming units at 90 min and at the time before adding the peptides, respectively [50].

### 4.9. Statistical Analysis

SPSS Statistics 20.0 for Windows (SPSS Inc. Chicago, IL, USA) was used for all statistical analyses. All data are given as mean ± SD unless otherwise indicated. A *t*-test was calculated to analyze the difference between the time killing and hemolysis activity of peptides. *p* values of <0.0001 were considered statistically significant.

## 5. Conclusions

In conclusion, the researcher investigated the structure–activity relationship of MpVT and its analog through the effect of amino acid substitutions and deletion of the first three C-terminal residues. The study showed that the first three amino acid deletions reduced the length of the α-helix structure, which decreased antibacterial activity. Furthermore, the amino acid replacement by bulky amino acid may be crucial to interaction.

## Figures and Tables

**Figure 1 molecules-27-00561-f001:**
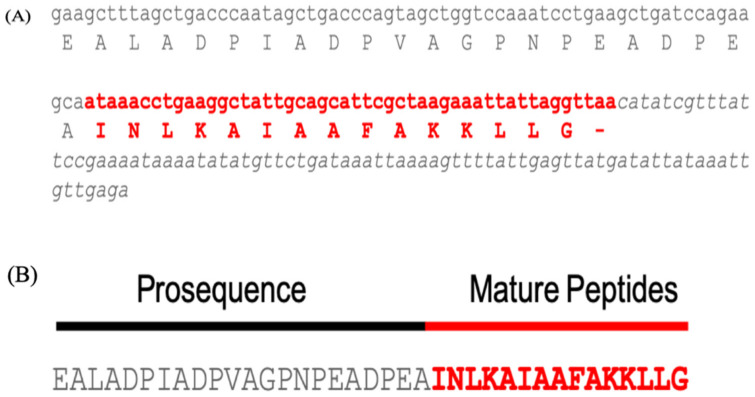
The partial nucleotide sequence and deduced amino acid sequence of *V. tropica* mastoparans (MpVT): (**A**) The nucleotide sequence of MpVT. The stop codon of the sequences marks as -. The mature sequences are shown in red letters and italic type, which were obtained by Edman degradation. (**B**) The deduced MpVT amino acid sequences of prosequences and mature sequences.

**Figure 2 molecules-27-00561-f002:**
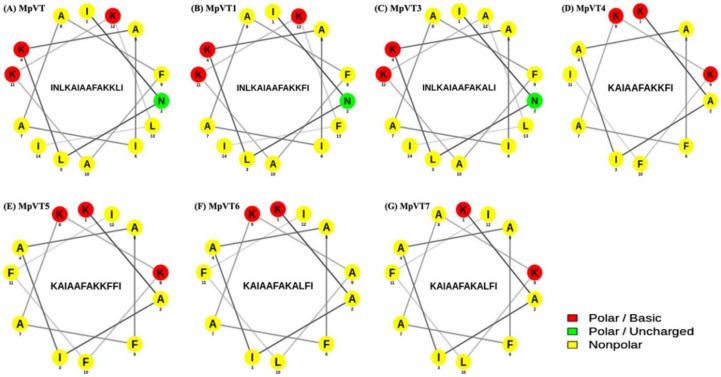
Helical wheel diagram of MpVT and its analogs (**A**–**G**). Helical wheel projection was performed using the Netwheels: Peptides Helical Wheel and Net projections maker. Red, green, and yellow circles represent polar/basic amino acids, polar/uncharged amino acids, and non-polar amino acids, respectively.

**Figure 3 molecules-27-00561-f003:**
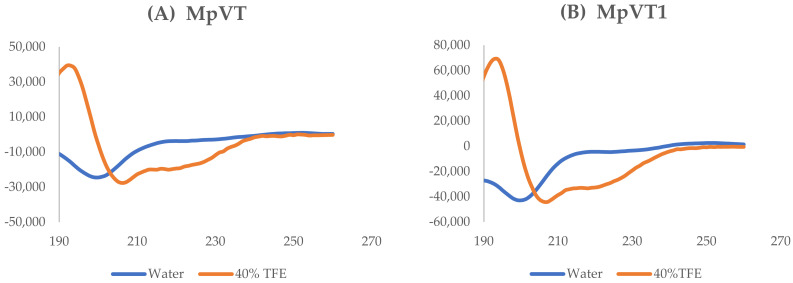
Circular dichroism (CD) spectra of MpVT and its analogs obtained at 100 μg/mL peptide concentration at 25 °C (**A**–**G**). All MpVTs showed unordered conformation in water (blue line) and characteristic α-helical spectra in the presence of 40% TFE (red line).

**Figure 4 molecules-27-00561-f004:**
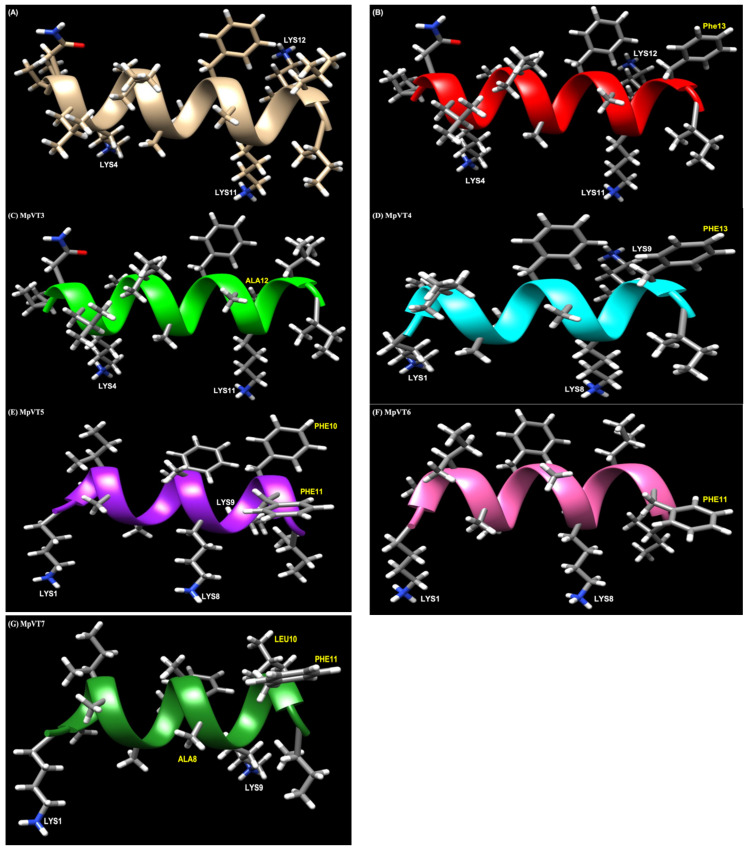
Structural models of MpVT and its analogs: (**A**) MpVT; (**B**) MpVT1; (**C**) MpVT3; (**D**) MpVT4; (**E**) MpVT5; (**F**) MpVT6; and (**G**) MpVT7. The models present the secondary structure in ribbon structure and the amino acid in sticky residues. Cationic residues (Lys amino acid) are labeled in white letters and the amino acid substitution is labeled in yellow letters.

**Figure 5 molecules-27-00561-f005:**
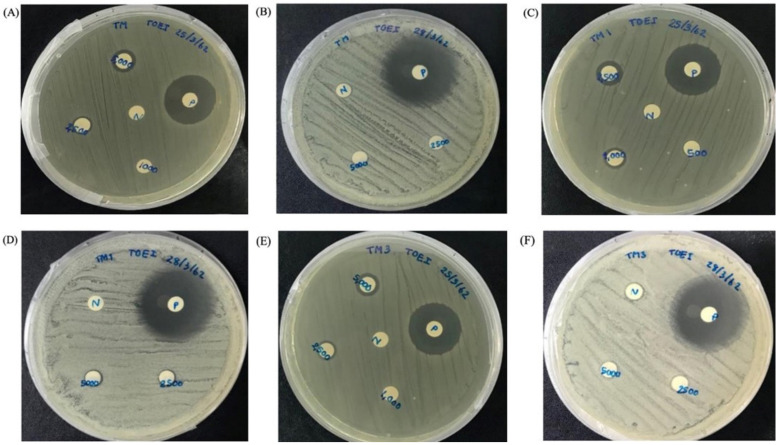
Antibacterial activities assay of *E. coli* (0157:H7) and *S. aureus* (ATCC 25923) treated with MpVT (**A**,**B**), MpVT1 (**C**,**D**), and MpVT3 (**E**,**F**) by disk diffusion method.

**Figure 6 molecules-27-00561-f006:**
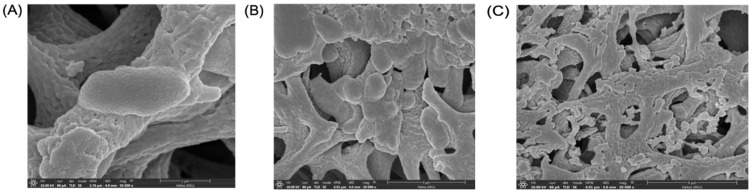
Scanning electron micrographs of *E. coli* treated with peptides: (**A**) control bacteria after treatment with 0.01% acetic acid for 1 h; (**B**,**C**) bacteria after treatment with MpVT at 2× MIC for 1 h.

**Figure 7 molecules-27-00561-f007:**
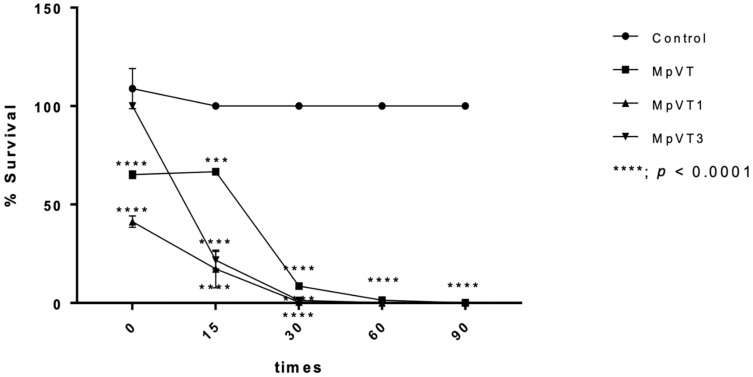
Time–kill kinetics of MpVT, MpVT1, MpVT3 and Control against *E. coli* 0157:H7 (open triangles) at a concentration two-fold above the MIC (100 μg/mL). Controls correspond to bacteria incubated in PBS without peptide. Data are the means ± S.E.M. of one experiment performed in triplicate; *** *p* < 0.001 versus control, **** *p* < 0.0001 versus control.

**Figure 8 molecules-27-00561-f008:**
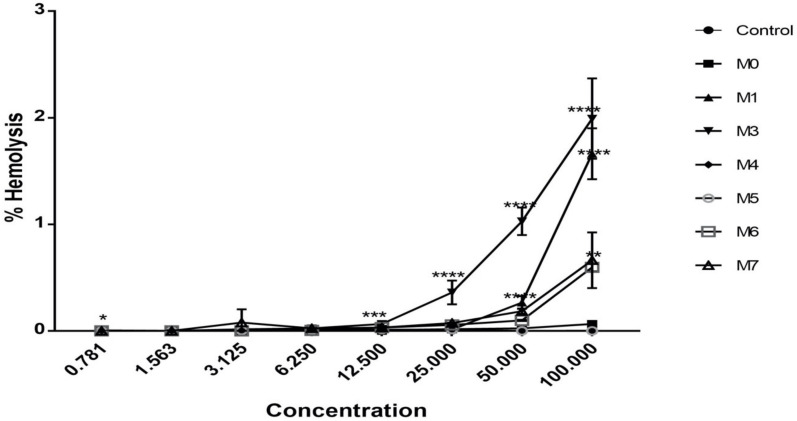
Hemolytic activity of MpVT and its analogs on human erythrocytes. Triton-X 100 was used as a positive control. These concentrations represent the mean values of three independent experiments performed in duplicate. Data are the means ± S.E.M. of one experiment performed in triplicate; * *p* < 0.3 versus control, ** *p* < 0.02 versus control, *** *p* < 0.07 versus control. **** *p* < 0.0001 versus control.

**Table 1 molecules-27-00561-t001:** The physicochemical prediction of the peptides.

Peptides	Sequences	AA	Molecular Weight (Da)	Charge	%H	H	μH
MpVT	INLKAIAAFAKKLI	14	1513.95	+3	71.43	0.590	0.403
MpVT1	INLKAIAAFAKKFI	14	1547.96	+3	71.43	0.596	0.409
MpVT3	INLKAIAAFAKALI	14	1456.85	+2	78.57	0.683	0.361
MpVT4	KAIAAFAKKFI	11	1207.54	+3	72.73	0.495	0.611
MpVT5	KAIAAFAKKFFI	12	1354.71	+3	75.00	0.603	0.411
MpVT6	KAIAAFAKALFI	12	1263.60	+2	83.33	0.704	0.370
MpVT7	KAIAAFAAKLFI	12	1263.60	+2	83.33	0.704	0.321

The sequences of MpVT were obtained from N-terminal sequence. The red letters indicate that the amino acid had changed; the blue letters indicate an added amino acid; %H, hydrophobic residues; H, hydrophobicity; μH, hydrophobic moment; molecular weight of peptides observed by mass spectrophotometry from Genic bio Inc. (Shanghai, China). The charge, hydrophobicity, and hydrophobic moment were calculated using Heliquest (https://heliquest.ipmc.cnrs.fr/cgi-bin/ComputParams.py, accessed on 25 June 2021).

**Table 2 molecules-27-00561-t002:** Calculated percentage of helicity for MpVT and its analogs under differences conditions ^a^.

Peptides	% Content in Water		% Content in 40% TFE	
	α-Helix	β-Sheet	α-Helix	β-Sheet
MpVT	11.67	4.74	95.22	0.02
MpVT1	8.96	6.6	95.22	0.02
MpVT3	8.96	6.39	95.34	0.02
MpVT4	2.04	9.36	95.16	0.01
MpVT5	2.04	9.95	95.23	0.01
MpVT6	1.51	9.38	94.64	0.01
MpVT7	1.51	8.95	95.50	0.01

^a^ The percentage content in α-helix and β-sheet of the peptides was estimated by the K2D3 method (http://cbdm-01.zdv.uni-mainz.de/~andrade/k2d3/, accessed on 19 October 2021) with an estimated maximum error of >0.32.

**Table 3 molecules-27-00561-t003:** Antimicrobial activity and hemolysis activity of MpVT and its analogs.

Microorganism	MIC (μg/mL)						
	MpVT	MpVT1	MpVT3	MpVT4	MpVT5	MpVT6	MpVT7
*P. aeruginosa* ATCC 27,853 (DMST 4739)	25.00	6.25	6.25	25.0	>50.00	>50.00	>50.00
*S. aureus* ATCC25923	50.00	12.50	50.00	>50.00	>50.00	>50.00	>50.00
*B. subtilis* ATCC663	25.00	3.125	12.50	>50.00	>50.00	>50.00	50.00
*E. coli* 0157:H7	50.00	6.25	50.00	>50.00	>50.00	>50.00	>50.00
*E. coli* DH5α	3.125	0.39	0.78	25.00	6.25	12.50	6.25
*K. pneumoniae* ATCC27736 (DMST4739)	25.00	6.25	25.00	>50.00	>50.00	>50.00	>50.00

MIC, minimum inhibitory concentration; the three independent experiments were performed in triplicate.

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
