# Peer review of "In Silico and In Vitro Structure–Activity Relationship of Mastoparan and Its Analogs"

_molecules, 2022, doi:10.3390/molecules27020561_

Round 1
Reviewer 1 Report
In silico and in vitro structure-activity relationship of mastoparan and its analogues were investigated. In present research, the effect of amino acid substitutions and deletion of the first three C-terminal residues on the structure-activity relationship was investigated.
The secondary structure and the biological activity of the seven designed analogues were explored. The biological activity assays showed that the substitution of phenylalanine (MpVT1) resulted in a higher antibacterial activity in comparison to MpVT. The analogues with the first three C-terminal residues deleted, showed decreased biological activity. The CD spectra of these peptides showed a high content α-helical conformation in the presence of 40% 2,2,2- trifluoroethanol (TFE). The first three C-terminal deletions reduced the length of the α-helix explaining decreased biological activity. MpVTs showed that the hemolytic activity of mastoparan was correlated to mean hydrophobicity and mean hydrophobic moment.
The manuscript is well designed, the research was performed carefully applying adequate methods and the result are presented adequately. Therefore, I can recommend the acceptance of the present manuscript after small correction indicated further.
Although at the end of introduction part, it is written what was done in present work, there is need to point out the novelty of present research. What was done for the first time? What is the hypothesis of present research?
Author Response
Response 1: We really appreciate your efforts to review the articles. The introduction of the manuscript was changed.
“Mastoparan is a 14-amino acid, amphipathic and cationic peptide obtained from wasp venom. They exhibit several biological activities including mast cell degranulation, hemolysis and the release of histamines. Most of mastoparan sequences, there have many hydrophobic amino acids and 2-4 Lys residues. The majority of hydrophobic nature amino acids in mastoparans peptides leads them to adopt amphipathic a-helix conformation in micelle solvents and membrane-bound state. The physicochemical properties of the peptides, such as the length, charge distribution, net charge, molecular volume, amphipathicity and oligomeric state in solution, play essential roles in the interactions between the peptide and the membrane surface and/or with the membrane core. The wasp venom mastoparan represents a potential source of pharmacology substances AMPs. However, some of the wasp venom mastoparan are not optimal for therapeutic applications due to limitations such as toxicity, salt sensitivity, and bioavailability, which limit their evolution into therapeutically effective antibiotics. In this regard, various approaches have been employed to optimize the original sequence of AMPs, including residue substitution and/or truncation.

Reviewer 2 Report
In the current study, the authors have evalauted "In Silico and In Vitro Structure-Activity Relationship of Masto- 2 paran and its Analogues" the authors have investigated very important study which is interesting for the readers and investigators. However, they need to improve it before publication. The following minor points should be considered.
Abstract: gram positive must Gram.
The abbreviations have been repeated consistently in the abstract section. It is enough to define first time and then use the abbreviations.
The conclusion of the abstract has not been well summarized in term of the results obtained.
Introduction.
Line 58: ......was influenced???
In my opinion, mastoparan has not been well introduced in the introduciton section. The real picture of doing this work is very obscured. THe author should focus and extend with me review of literature, what were the real motives behind this work. The authors should should show the shortcomings of the previous work and the present investigation a link to be established.
Authors should show the materials and methods section immedialty after the introduction or as suggested by the respective journal.
What about the statistical analysis of this work?
Why the antimicrobial activity were registered against a few bacteria The authors should also investigate other pathogenic bacteria. After that they can conclude their present results. On the basis of a few bacteria, how can you give your conclusion. Altenatively, you have to show in the conclusion only the specific bacteria.
I do not see any statistical analysis on the most of the parameters which is must.
The discussion is mere a review of literature and does not discuss the findings well.
Author Response
Point 1: Abstract: gram positive must Gram. The abbreviations have been repeated consistently in the abstract section. It is enough to define first time and then use the abbreviations.
Response 1: Thank you for your contribution to spend your valuable time to review our manuscript. For abstract, gram positive has been changed to Gram. The abbreviations have been changed.
Point 2: The conclusion of the abstract has not been well summarized in term of the results obtained.
Response 2: The abstract has been changed
Point 3: Introduction. Line 58: ......was influenced??? In my opinion, mastoparan has not been well introduced in the introduction section. The real picture of doing this work is very obscured. The author should focus and extend with me review of literature, what were the real motives behind this work. The authors should show the shortcomings of the previous work and the present investigation a link to be established.
Response 3:
Line 58: “the phytochemical properties” have been changed to “physiochemical properties.”
The introduction of manuscript was changed.
“Mastoparan is a 14-amino acid, amphipathic and cationic peptide obtained from wasp venom. They exhibit several biological activities including mast cell degranulation, hemolysis and the release of histamines. Most of mastoparan sequences, there have many hydrophobic amino acids and 2-4 Lys residues. The majority of hydrophobic nature amino acids in mastoparans peptides leads them to adopt amphipathic a-helix conformation in micelle solvents and membrane-bound state. The physicochemical properties of the peptides, such as the length, charge distribution, net charge, molecular volume, amphipathicity and oligomeric state in solution, play essential roles in the interactions between the peptide and the membrane surface and/or with the membrane core. The wasp venom mastoparan represents a potential source of pharmacology substances AMPs. However, some of the wasp venom mastoparan are not optimal for therapeutic applications due to limitations such as toxicity, salt sensitivity, and bioavailability, which limit their evolution into therapeutically effective antibiotics. In this regard, various approaches have been employed to optimize the original sequence of AMPs, including residue substitution and/or truncation.
Point 4: Authors should show the materials and methods section immediately after the introduction or as suggested by the respective journal.
Response 4: The materials and methods section has been followed by the respective journal
Point 5: What about the statistical analysis of this work?
Response 5: Statistical analysis was added into material and method 4.9; “SPSS Statistics 20.0 for Windows (SPSS Inc. Chicago, IL, USA) was used for all statistical analyses. All data are given as mean ± SD unless otherwise indicated. A t-test was calculated to analyze the difference between time killing and hemolysis activity of peptides. p values of < 0.0001 were considered statistically significant”.
Point 6: Why the antimicrobial activity were registered against a few bacteria. The authors should also investigate other pathogenic bacteria. After that they can conclude their present results. On the basis of a few bacteria, how can you give your conclusion. Alternatively, you have to show in the conclusion only the specific bacteria. I do not see any statistical analysis on the most of the parameters which is must. The discussion is mere a review of literature and does not discuss the findings well.
Response 6: The comment is very sharp. However, this manuscript was studied in terms of structure relationship of the peptides and its analogue. The structure analysis was used to analysis the peptides functional correlated to the biological activity. Both gram of bacteria of Gram-positive and Gram-negative bacteria had been representatively checked. However, many of pathogenic bacteria have been future investigated.
